# EBF1 regulates sensory establishment in the cochlea by positioning the medial boundary of the prosensory domain and restricting proliferation of the sensory progenitor population

Kathryn G. Powers[1,2], Joshua Hahn[1], Juliette Wohlschlegel[1] and Olivia Bermingham-McDonogh[1,*]

## ABSTRACT

In our previous study, we reported that *Ebf1* excision throughout the inner ear epithelium and before the onset of cochlear development leads to dramatic sensory expansion in the cochlea by neonatal stages. *Ebf1* conditional knockout cochleae possess over twice as many sensory cells as littermate controls and develop ectopic sensory patches in their Kölliker's organs. To better understand the mechanism behind the role of EBF1 in restricting sensory establishment, we performed multiome sequencing in our current study. EBF1 is a transcription factor best known for its importance in B cell lineage specification, during which it acts as both an activator and a repressor. Our results indicate that in mice EBF1 prevents the Kölliker's organ cells from being recruited to the prosensory domain by promoting expression of *Prdm16* and repressing expression of *Jag1* and *Sox2*. We also found that EBF1 may promote cell cycle exit by repressing *Ccnjl* expression. In summary, medial expansion of the prosensory domain, together with delayed cell cycle exit in the developing cochlear epithelium, underlies the robust increase in sensory cells seen in *Ebf1* conditional knockouts.

KEY WORDS: Cochlea, Sensory, Multiome, EBF1, Inner ear, Cell fate, Mouse

## INTRODUCTION

The sensory region of the cochlea, called the organ of Corti, contains both hair cells (HCs) and support cells (SCs). This region develops from an unpatterned epithelium through a process dependent on BMP, FGF, WNT, Hh and Notch signaling (reviewed by Brown and Groves, 2020; Ebeid and Huh, 2017; Hayashi et al., 2007, 2008b; Mansour and Urness, 2025; Munnamalai and Fekete, 2016; Ohyama et al., 2010; Riccomagno et al., 2002). A complex gradient of factors determines the medial and lateral borders of this sensory region. While the signals that establish the medial boundary of the prosensory domain are not well defined, BMP4 is thought to be crucial in specifying the lateral boundary (Ohyama et al., 2010). BMP4 is expressed in the outer sulcus and patterns the cochlea in a

dose-dependent manner. High BMP4 inhibits prosensory domain formation, whereas low BMP4 concentrations appear to be necessary for prosensory specification (Ohyama et al., 2010). Following cochlear patterning and sensory differentiation, the organ of Corti consists of one row of inner HCs (iHCs) and three rows of outer HCs (oHCs). Each iHC is surrounded by inner phalangeal cells, and each oHC has an associated Deiters' cell. In between the rows of iHCs and oHCs are two rows of pillar cells (inner and outer) that differentiate with prominent microtubule bundles crosslinked with actin to form the tunnel of Corti (Fig. 1; Tolomeo and Holley, 1997; Tucker et al., 1999). The sensory epithelium (SE) lies above the basilar membrane which vibrates in response to sound, allowing the stereocilia on the tops of the HCs to deflect and thereby initiate the mechanotransduction that leads to neurotransmitter release from the iHCs. The signal is then relayed to the brain by the spiral ganglion neurons that synapse onto the HCs.

We first became interested in early B cell factors (EBFs) following a bulk ATAC sequencing study. In addition to detecting enrichment of motifs associated with transcription factors already known to be important in cochlear development, we identified EBF-binding motifs as being enriched in the open chromatin of cochlear prosensory and sensory cells (Wilkerson et al., 2019). A complementary single cell RNA sequencing study revealed that *Ebf1* is strongly expressed in the developing cochlear SE while the other members of the Ebf family, *Ebf2-Ebf4*, show little to no expression (Powers et al., 2024). Thus, we chose to look at cochlear development in a conditional knockout (cKO) mouse for *Ebf1*.

EBFs contain an unusual zinc-finger DNA binding domain, an IPT/TIG domain, a helix-loop-helix domain and a transactivation region (Hagman et al., 1995). They typically form stable dimers via the HLH domain and bind a palindromic consensus DNA motif (5′-TCCCNNGGGA-3′; Hagman et al., 1993; Travis et al., 1993) to regulate cell fate decisions, differentiation and migration (reviewed by Liberg et al., 2002). EBF1 is the most extensively studied member of the EBF family, and the transcription factor has well-documented roles in a diverse range of developmental processes, including adipocyte differentiation (Jimenez et al., 2007) and neurogenesis (Catela et al., 2019; Davis and Reed, 1996; Garel et al., 1997; Lobo et al., 2008; Wang and Reed, 1993; Wang et al., 1997). Notably, EBF1 (also known as OLF1), is important in olfactory development, where it regulates the transition from differentiation to maturation in olfactory receptor neurons (Cheng and Reed, 2007; Roby et al., 2012). As its name implies, however, EBF1 is best known for its roles in B cell progenitor proliferation and differentiation, during which it acts as both an activator and repressor (Li et al., 2018; reviewed by Ramírez et al., 2010; Treiber et al., 2010). EBF1 is required for activation of *Pax5* expression (Decker et al., 2009), and the two transcription factors are thought to work together to promote B cell specification by downregulating Notch signaling and repressing

[1]Department of Neurobiology and Biophysics, University of Washington Medical School, Seattle, WA 98195, USA. [2]Graduate Program in Molecular and Cellular Biology, University of Washington, Seattle, WA 98195, USA.

*Author for correspondence (oliviab@uw.edu)

K.G.P., 0000-0003-3278-9766; J.H., 0000-0002-4776-2067; J.W., 0000-0003-4317-9066; O.B., 0000-0002-2559-4218

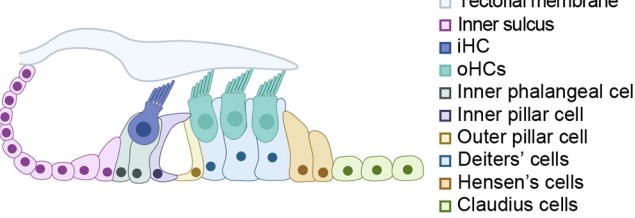

☐ Tectorial membrane
☐ Inner sulcus
■ iHC
☐ oHCs
☐ Inner phalangeal cells
☐ Inner pillar cell
☐ Outer pillar cell
☐ Deiters' cells
☐ Hensen's cells
☐ Claudius cells

**Fig. 1. The arrangement of hair cell and support cell subtypes within the mature organ of Corti.** The organ of Corti is composed of inner hair cells (iHCs) and outer hair cells (oHCs) along with multiple support cell subtypes, including inner phalangeal cells, pillar cells, Deiters' cells and Hensen's cells; just lateral to these lie the Claudius cells, which form the lateral boundary of the sensory epithelium. Hair cells have stereocilia bundles on their apical surfaces. Unlike inner hair cells, the stereocilia bundles of outer hair cells are embedded in the overlying tectorial membrane. Created in BioRender by Powers, K, 2026. https://BioRender.com/bddrcfo. This figure was sublicensed under CC-BY 4.0 terms.

genes that specify alternative hematopoietic cell fates (Pongubala et al., 2008; Souabni et al., 2002; Thal et al., 2009). EBF1 also facilitates the transition from progenitor to B-cell-lineage-committed accessible chromatin by recruiting BRG1 (SMARCA4), the catalytic subunit of the SWI/SNF (BAF) chromatin remodeling complex, via its C-terminal transactivation domain (Wang et al., 2020; Zolotarev et al., 2022). EBF1 integrates activation, repression and chromatin remodeling to ensure proper B cell programming.

We previously reported the cochlear phenotype of an *Ebf1*-cKO mouse, which uses the *Slc26a9* promoter to direct Cre-mediated excision of *Ebf1* throughout the inner ear epithelium at the otocyst stage. In the developing mouse cochlea, *Ebf1* is strongly expressed in both the epithelium and the surrounding mesenchyme. Briefly, the *Ebf1*-cKO phenotype involves an increase in early proliferation of the precursor cells such that the SE is doubled in size. We also saw a further increase in iHCs, suggesting that EBF1 is also important in regulating the medial border of the developing organ of Corti (Powers et al., 2024). In this study, we advance our understanding of this mutant by investigating how EBF1 shapes both the transcription and open chromatin landscapes of cells within the developing cochlea. By characterizing EBF1-binding motif accessibility within differentially expressed genes in cell-type-matched populations in the *Ebf1*-cKO and control littermate datasets, we were able to identify the dual role of EBF1 as a transcriptional activator and repressor during crucial stages of cochlear development. Instead of the expected decrease in chromatin accessibility from lost BRG1 interactions, we observed an overall increase, which we attribute to the EBF1 indirect upregulation of *Sox2*. Within the Kölliker's organ, the transient epithelial structure that gives rise to the inner sulcus of the cochlea, EBF1 appears to be required to promote *Prdm16* expression and repress *Jag1* and *Sox2* expression, and thus prevent Kölliker's organ cells from taking on a prosensory fate. We also found that EBF1 may directly regulate cell cycle exit in the developing cochlear epithelium by repressing *Ccnj1* expression. Loss of EBF1 interactions with these transcriptional targets leads to prolonged proliferation that is accompanied by an increase in the size of the sensory progenitor pool.

## RESULTS
### Ebf1 excision leads a medial shift in the boundary separating the prosensory domain from Kölliker's organ as early as E14.5
We previously reported that, at neonatal stages, *Ebf1*-cKO cochleae exhibit dramatic sensory expansion, with more than twofold increases in HCs and SCs within the differentiated SE as well as ectopic

sensory patches within the Kölliker's organ (Powers et al., 2024). To investigate whether changes in prosensory domain establishment contribute to this expansion, immunostaining was performed at embryonic day (E) 14.5, a developmental stage when the cochlear floor is patterned along the medial-lateral axis into the Kölliker's organ, prosensory domain and outer sulcus (Ohyama et al., 2010). Because *Ebf1* is strongly expressed in the developing cochlear epithelium, while *Ebf2-Ebf4* show little to no expression (Powers et al., 2024), a pan-EBF antibody was used to label EBF1. Anti-SOX2 and anti-PRDM16 antibodies were used to label the prosensory domain and Kölliker's organ, respectively. SOX2 is a transcription factor necessary for sensory development in the cochlea (Dabdoub et al., 2008; Kiernan et al., 2005b) and PRDM16 is a transcription factor important in the development of the Kölliker's organ (Ebeid et al., 2022; Zhang et al., 2025). At E14.5, EBF1 was strongly expressed in the PRDM16$^+$ Kölliker's organ and weakly expressed in the SOX2$^{high+}$ prosensory domain of control cochleae. By contrast, *Ebf1*-cKO littermate cochleae exhibited a much smaller PRDM16$^+$ Kölliker's organ and a much larger SOX2$^{high+}$ prosensory domain (Fig. 2A). Interestingly, the few PRDM16$^+$ Kölliker's organ cells that remained in the *Ebf1*-cKO cochleae appeared to show weak EBF labeling. This Ebf expression could be the result of residual *Ebf1* expression or from *Ebf3* and *Ebf4*, which show modest levels of expression in the Kölliker's organ at this developmental time point (Powers et al., 2024).

To enable investigation of EBF1-dependent changes in cochlear patterning at the single cell level, multiome sequencing, which combines single nucleus RNA sequencing (snRNA-seq) and single nucleus ATAC sequencing (snATAC-seq), was performed at E14.5. Multiome sequencing makes it possible to examine shifts in the distribution of cell types across the entire cochlear epithelium and to control for patterning differences caused by delayed convergent extension in *Ebf1*-cKO embryonic cochlear ducts (Powers et al., 2024). The multiome datasets were generated from nuclei isolated from six control and six *Ebf1*-cKO cochlear ducts collected from one litter of four control and four *Ebf1*-cKO mice. The epithelial cells were separated from the non-epithelial cells (Fig. S1) and clustering was performed on the epithelial cells using the control and *Ebf1*-cKO snRNA-seq data (Fig. S2). As expected, snATAC-seq analysis of the *Ebf1*-cKO relative to control cochleae showed reduced predicted EBF1-binding activity in their Kölliker's organ cells and prosensory cells (Fig. S3A). Surprisingly, however, snRNA-seq *Ebf1* transcript levels appeared to be comparable between *Ebf1*-cKO and control cochleae (Fig. S3B). This result can be attributed to both a poly-A stretch at the transcription start site of *Ebf1* transcripts that enables capture via poly(dT) priming and the design of the *Ebf1*-floxed allele (Vilagos et al., 2012). Exons 6-16 of *Ebf1* were excised in *Ebf1*-cKOs, leaving only the first 14 codons of exon 6, the exon that encodes the EBF1 zinc coordination motif, linked in frame to *Gfp*. While coverage near the *Ebf1* transcription start site was similar for *Ebf1*-cKO and control samples (Fig. S3C), coverage in exons 6-16 was significantly reduced in the *Ebf1*-cKO (Fig. S3D), consistent with their successful excision in the *Ebf1*-cKOs.

The distribution of epithelial cells in *Ebf1*-cKO and control snRNA-seq uniform manifold approximation and projections (UMAPs) (Fig. 2B,C) recapitulated the phenotype seen in the stage-matched embryonic immunostaining experiment (Fig. 2A). Like *Sox2*, *Hey2* is expressed by prosensory cells (Hayashi et al., 2008a); however, *Hey2* expression was more restricted to just the prosensory domain (Fig. S2). In the E14.5 control cochleae, the *Prdm16*$^{high+}$ Kölliker's organ accounted for 16.9% of epithelial cells while the *Hey2*$^+$ prosensory cells represented 21.4%. In E14.5

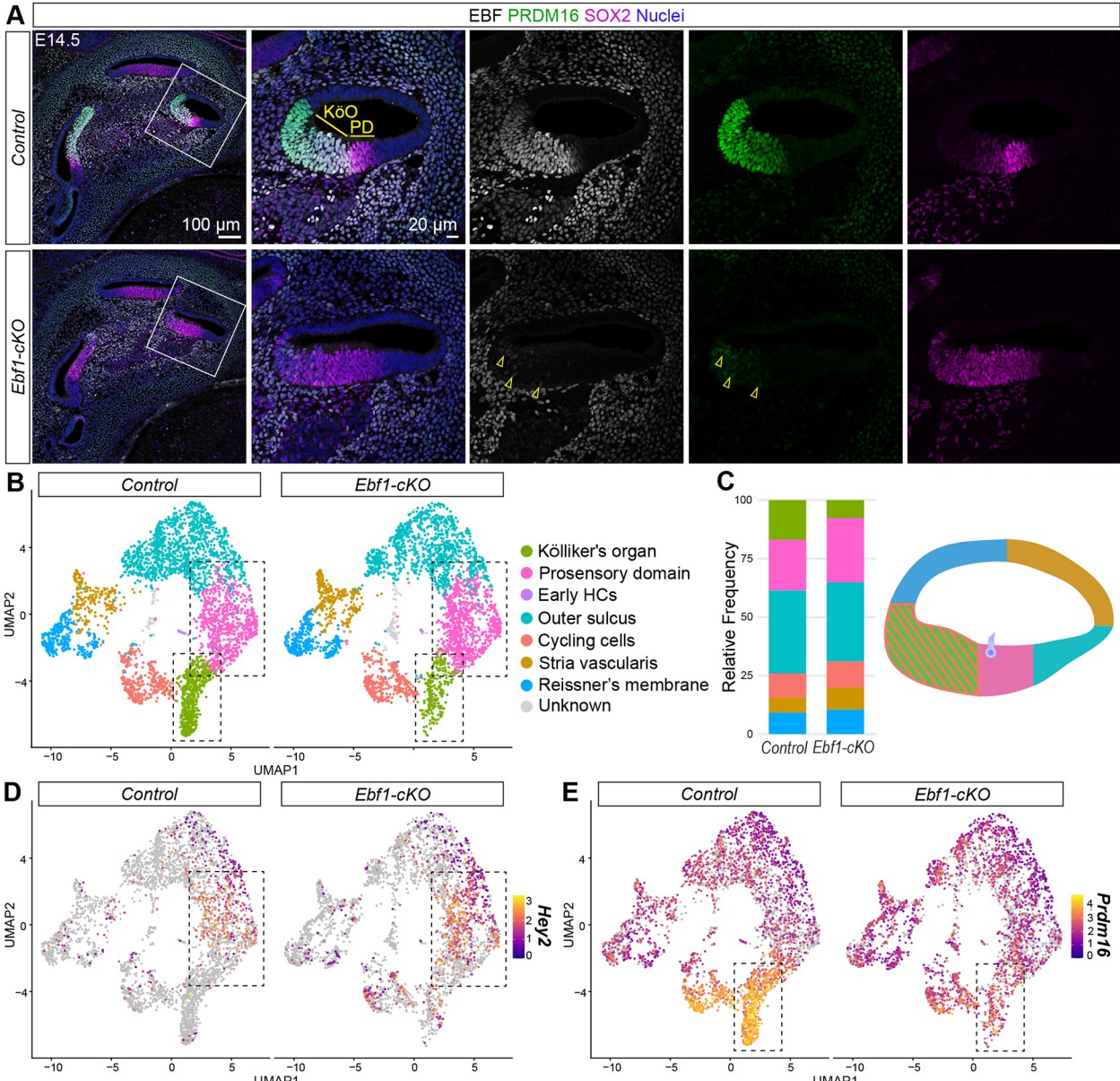

**Fig. 2. Immunostaining and snRNA-seq at E14.5 reveal the co-option of Kölliker's organ cells into the *Ebf1*-cKO prosensory domain.** (A) Confocal images of E14.5 mid-modiolar sections from control and *Ebf1*-cKO littermates capture an EBF1-dependent expansion of the SOX2high+ prosensory domain (PD) and a reduction of the PRDM16+ Kölliker's organ (KöO). White boxes (left) indicate regions included in the zoomed-in images (right). Yellow arrowheads mark example nuclei positive for both EBF and PRDM16 in the *Ebf1*-cKO sample. (B) E14.5 snRNA-seq UMAP plots of single nuclei from the cochlear epithelium, separated by genotype. Canonical markers were used to assign cell types to nuclei clusters. HCs, hair cells. (C) Relative frequencies of cell types identified in the control (16.9% Kölliker's organ, 21.4% prosensory domain, 0.3% early HCs, 35.4% outer sulcus, 10.3% cycling cells, 6.5% stria vascularis and 9.2% Reissner's membrane) and the *Ebf1*-cKO (7.6% Kölliker's organ, 27.3% prosensory domain, 0.2% early HCs, 33.7% outer sulcus, 11.2% cycling cells, 9.4% stria vascularis and 10.5% Reissner's membrane) datasets. Cell types in the frequency plot are color-coded to match their assigned colors in both the UMAPs and the cross-sectional diagram of an E14.5 control cochlea. Created in BioRender by Powers, K, 2026. https://BioRender.com/786qacu. This figure was sublicensed under CC-BY 4.0 terms. (D,E) E14.5 snRNA-seq feature plots split by genotype highlight *Hey2* expression in the prosensory domain (D) and strong *Prdm16* expression in the Kölliker's organ (E). Dashed boxes indicate regions within the feature plots that contain nuclei assigned to the prosensory domain and Kölliker's organ.

*Ebf1*-cKO cochleae, the relative number of Kölliker's organ cells dropped by more than half to 7.6% and prosensory cells increased to 27.3%, while the percentages for the other epithelial cell types remained largely unchanged compared with the littermate controls (Fig. 2C-E). In summary, both the E14.5 immunostaining and multiome sequencing experiments demonstrate that loss of EBF1 during cochlear patterning leads to the co-option of Kölliker's organ cells into the prosensory domain.

**Ebf1 excision leads to delayed cell cycle exit as late as E16.5 in the developing cochlear epithelium**

In the developing cochlea, the prosensory domain is defined between E12 and E15 as prosensory cells undergo cell cycle exit in a wave that starts at the apex and progresses towards the base (Chen and Segil, 1999; Ruben, 1967). These prosensory cells subsequently differentiate into HCs and SCs in a wave that proceeds in the opposite direction, starting at the base at ~E14 and reaching

the apex during neonatal stages (Chen et al., 2002). To assess how the changes in cochlear patterning observed at E14.5 impact development at later stages when HC and SC differentiation is well underway, immunostaining was performed at E16.5. Much like at E14.5, EBF1 is strongly expressed in the Kölliker's organ and weakly expressed in the developing SE, and the expanded *Ebf1*-cKO SOX2$^{high+}$ developing SE is adjacent to a smaller PRDM16$^+$ Kölliker's organ (Fig. 3A). Interestingly, the *Ebf1*-cKO PRDM16 expression domain appeared to be less diminished at this

later stage, suggesting that compensatory pathways may begin to counteract loss of *Ebf1* by E16.5 and allow for the Kölliker's organ to reestablish by neonatal stages (Powers et al., 2024).

Multiome sequencing was performed at E16.5 to investigate how loss of *Ebf1* impacts cochlear patterning, cell cycle exit and differentiation at the single cell level. The E16.5 multiome datasets were generated from six control and six *Ebf1*-cKO cochlear ducts from one litter of four control and three *Ebf1*-cKO mice. *Ebf1*-cKO and control epithelial cells were separated from the non-epithelial

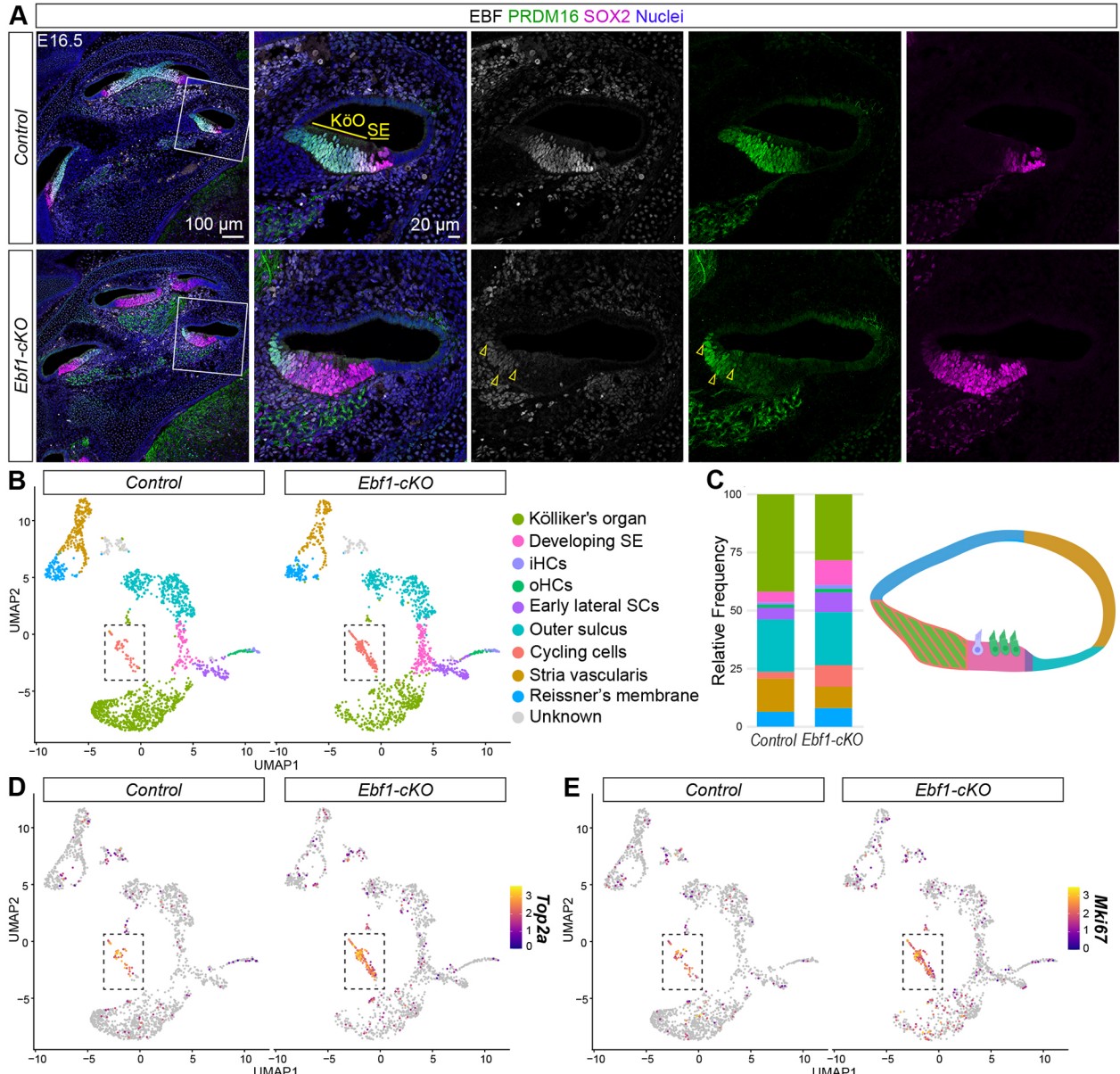

Fig. 3. Immunostaining and snRNA-seq at E16.5 capture delayed cell cycle exit in *Ebf1*-cKOs. (A) Confocal images of E16.5 mid-modiolar sections from control and *Ebf1*-cKO littermates capture an EBF1-dependent expansion of the SOX2$^{high+}$ sensory epithelium (SE) and a reduction of the PRDM16$^+$ Kölliker's organ (KöO). White boxes (left) indicate regions included in the zoomed-in images (right). Yellow arrowheads mark example nuclei positive for both EBF and PRDM16 in the *Ebf1*-cKO sample. (B) E16.5 snRNA-seq UMAPs of single nuclei from the cochlear epithelium, separated by genotype. Canonical markers were used to assign cell types to nuclei clusters. iHCs, inner hair cells; oHCs, outer hair cells, SCs, support cells. (C) Relative frequencies of cell types identified in the control (41.9% Kölliker's organ, 4.6% developing SE, 1.0% iHCs, 1.3% oHCs, 5.0% early lateral SCs, 22.5% outer sulcus, 3.0% cycling cells, 14.3% stria vascularis and 6.4% Reissner's membrane) and the *Ebf1*-cKO (28.3% Kölliker's organ, 10.6% developing SE, 1.7% iHCs, 1.5% oHCs, 8.5% early lateral SCs, 22.8% outer sulcus, 9.3% cycling cells, 9.2% stria vascularis and 8.1% Reissner's membrane) datasets. Cell types in the frequency plot are color-coded to match their assigned colors in both the UMAPs and the cross-sectional diagram of an E16.5 control cochlea. Created in BioRender by Powers, K, 2026. https://BioRender.com/r22p7mm. This figure was sublicensed under CC-BY 4.0 terms. (D,E) E16.5 snRNA-seq feature plots split by genotype highlight *Top2a* (D) and *Mik67* (E) expression in nuclei from cycling cells. Dashed boxes indicate regions within the feature plots that contain nuclei assigned to the cycling cells.

populations (Fig. S4), and clustering was performed on the epithelial subset using the snRNA-seq data (Fig. S5). In the E16.5 control cochleae, the Kölliker's organ accounted for 41.9% of epithelial cells while the developing SE, inner HCs (iHCs), outer HCs (oHCs) and early lateral SCs accounted for 11.9%. In E16.5 *Ebf1*-cKO cochleae, the relative number of Kölliker's organ cells dropped by about a third to 28.3% and the relative number of developing SE and differentiated sensory cells approximately doubled to 22.3% (Fig. 3B,C). In addition to capturing the recruitment of Kölliker's organ cells to the developing SE observed in the stage-matched embryonic immunostaining experiment, the E16.5 snRNA-seq data highlighted an increase in the relative number of $Top2a^+$ and $Mki67^+$ cycling cells in *Ebf1*-cKO compared with control cochleae (Fig. 3C-E). Cycling cells represented 3.0% of the control epithelial cells and 9.3% of the *Ebf1*-cKO epithelial cells (Fig. 3C). This result agrees with our previous finding that proliferation of *Ebf1*-cKO prosensory cells persists beyond E15 (Powers et al., 2024). Since the relative number of cycling cells at E14.5 is comparable between *Ebf1*-cKO and control cochleae (Fig. 2C), the increase detected at E16.5 is likely due to delayed cell cycle exit. EBF1 regulates sensory development in the cochlea by both positioning the medial boundary of the prosensory domain and restricting the proliferative capacity of the sensory progenitor population.

### EBF1 likely regulates Prdm16, Sox2 and Jag1 expression to prevent Kölliker's organ cells from adopting a prosensory cell fate

To investigate the mechanism underlying the role of EBF1 in cochlear patterning at E14.5, R packages Seurat and Signac were used to perform multiomic analyses involving both snRNA-seq and snATAC-seq data from *Ebf1*-cKO and control littermates. Differentially expressed genes in Kölliker's organ cells were identified to uncover potential EBF1 targets that explain the transcription factor's role in preventing their recruitment to the prosensory domain (Table S1). *Ebf1* excision led to an overall increase in chromatin accessibility in Kölliker's organ cells, with 510 ATAC peaks more accessible and only five peaks less accessible in *Ebf1*-cKOs relative to controls (Fig. 4A). Consistent with the E14.5 immunostaining results (Fig. 2A), *Prdm16* was among the 51 downregulated genes and *Sox2* was among the 106 upregulated genes in *Ebf1*-cKO compared with control Kölliker's organ cells (Fig. 4B). This shift is accompanied by upregulation of additional established prosensory markers, including *Hey2* and *Fgf20* (Hayashi et al., 2008a,b). Interestingly, *Jag1* expression was also elevated in *Ebf1*-cKO Kölliker's organ cells. JAG1 (jagged 1), a Notch ligand important in prosensory specification, is initially expressed throughout the cochlear floor at E12.5 and becomes restricted to the Kölliker's organ by E13.5 (Basch et al., 2016; Kiernan et al., 2006; Ohyama et al., 2010). JAG1 expression then shifts to the differentiating prosensory domain before ultimately being expressed in differentiated SCs by E16.5 (Basch et al., 2016; Murata et al., 2006). As JAG1 shifts from the Kölliker's organ to the prosensory domain, its expression domain becomes restricted to a column of cells along the medial boundary of the prosensory domain at E14.5. Loss of EBF1 led to a marked, medial expansion of the JAG1 expression domain (Fig. 4C,D). Like the dysregulation of *Prdm16* and *Sox2* expression, the upregulation of *Jag1* in the receding *Ebf1*-cKO Kölliker's organ suggests it may be a direct or indirect target of EBF1, essential for proper cochlear patterning.

After identifying potential EBF1 targets with the snRNA-seq data, the snATAC-seq data was used to look for evidence of direct regulation by EBF1. Differentially accessible chromatin regions within and 500 kb proximal to the gene loci were screened for both EBF1-binding motifs and a strong correlation with RNA expression using the 'LinkPeaks' function in Signac. Given that EBF1 is known to act as an activator as well as a repressor during B cell lineage specification (Li et al., 2018; reviewed by Ramírez et al., 2010; Treiber et al., 2010), the snRNA-seq and snATAC-seq data were analyzed together to determine the mode of regulation of EBF1 at each target locus. EBF1-dependent changes in chromatin accessibility were consistently more pronounced in Kölliker's organ cells than prosensory cells, and *Ebf1* excision in the cochlear epithelium caused the open chromatin of Kölliker's organ cells to take on a more prosensory-like pattern (Fig. 4E-G). Two open chromatin regions were identified that strongly correlate with *Prdm16* RNA expression and that contain EBF1 motifs, and both regions were less accessible in *Ebf1*-cKO than control Kölliker's organ cells (Fig. 4E). Taken together with our finding that *Ebf1* excision leads to downregulation of *Prdm16*, this decrease in chromatin accessibility suggests that EBF1 is needed to promote *Prdm16* expression in Kölliker's organ cells. Examination of open chromatin regions linked with *Jag1* RNA expression revealed regions containing EBF1 motifs that showed subtle changes and appeared to be more accessible in the *Ebf1*-cKO Kölliker's organ cells (Fig. 4F). These open chromatin regions also contained PRDM16 motifs. Given that *Ebf1*-cKOs show upregulation of *Jag1* in their Kölliker's organ cells, PRDM16 may interact with EBF1 to repress *Jag1* expression. Although we did not find any open chromatin regions near the *Sox2* locus that both contained EBF1 motifs and were linked to expression of the gene, we did detect a distal region (~500 kb from the transcription start site) containing a PRDM16 motif and associated with *Sox2* RNA expression (Fig. 4G). This region was more accessible in the *Ebf1*-cKO Kölliker's organ cells (Fig. 4G′). As *Ebf1* excision led to an upregulation of *Sox2* in Kölliker's organ cells, this finding suggests that PRDM16 directly represses *Sox2* expression in these cells. In summary, our analyses indicate that EBF1 is necessary to promote *Prdm16* expression and possibly to co-repress *Jag1* expression with PRDM16. PRDM16, in turn, appears to be necessary to repress *Sox2* expression. EBF1 regulation of these targets in the Kölliker's organ likely helps prevent the cells from assuming a prosensory identity.

Gene ontology (GO) term enrichment analysis using our snATAC-seq data revealed that the top two dysregulated terms in the Kölliker's organ cells of E14.5 *Ebf1*-cKOs compared with controls were 'negative regulation of Notch signaling' and 'regulation of Notch signaling' (Fig. S6A). Consistent with this finding, receptor-ligand analysis of the Notch pathway at E14.5 using CellChat showed that Notch signaling levels in *Ebf1*-cKO Kölliker's organ cells increased dramatically and resembled those seen in prosensory cells (Fig. S6B). This increase in Notch signaling is likely driven by the upregulation of *Jag1* observed in *Ebf1*-cKOs, as we did not find significant expression differences between *Ebf1*-cKO and control cochleae in other Notch-ligand genes (Fig. S6C) and our multiomic analyses indicated that EBF1 and PRDM16 co-repress *Jag1* expression.

### EBF1 may directly repress Ccnj1 expression to restrict cell proliferation in the developing cochlear epithelium

Loss of EBF1 leads to prolonged and expanded prosensory cell proliferation (Powers et al., 2024). At E16.5, proliferating cells in control cochleae are restricted to Kölliker's organ, whereas EdU$^+$ cells are seen in expanded SOX2$^{high+}$ developing SE in *Ebf1*-cKO cochleae (Fig. 5A). These cells can be seen throughout the expanded SE but are more concentrated in the medial half. To investigate the mechanism by which EBF1 regulates cell cycle exit, we examined E16.5 *Ebf1*-cKO and control littermates. We detected 71 upregulated and 27 downregulated cell-cycle related genes in

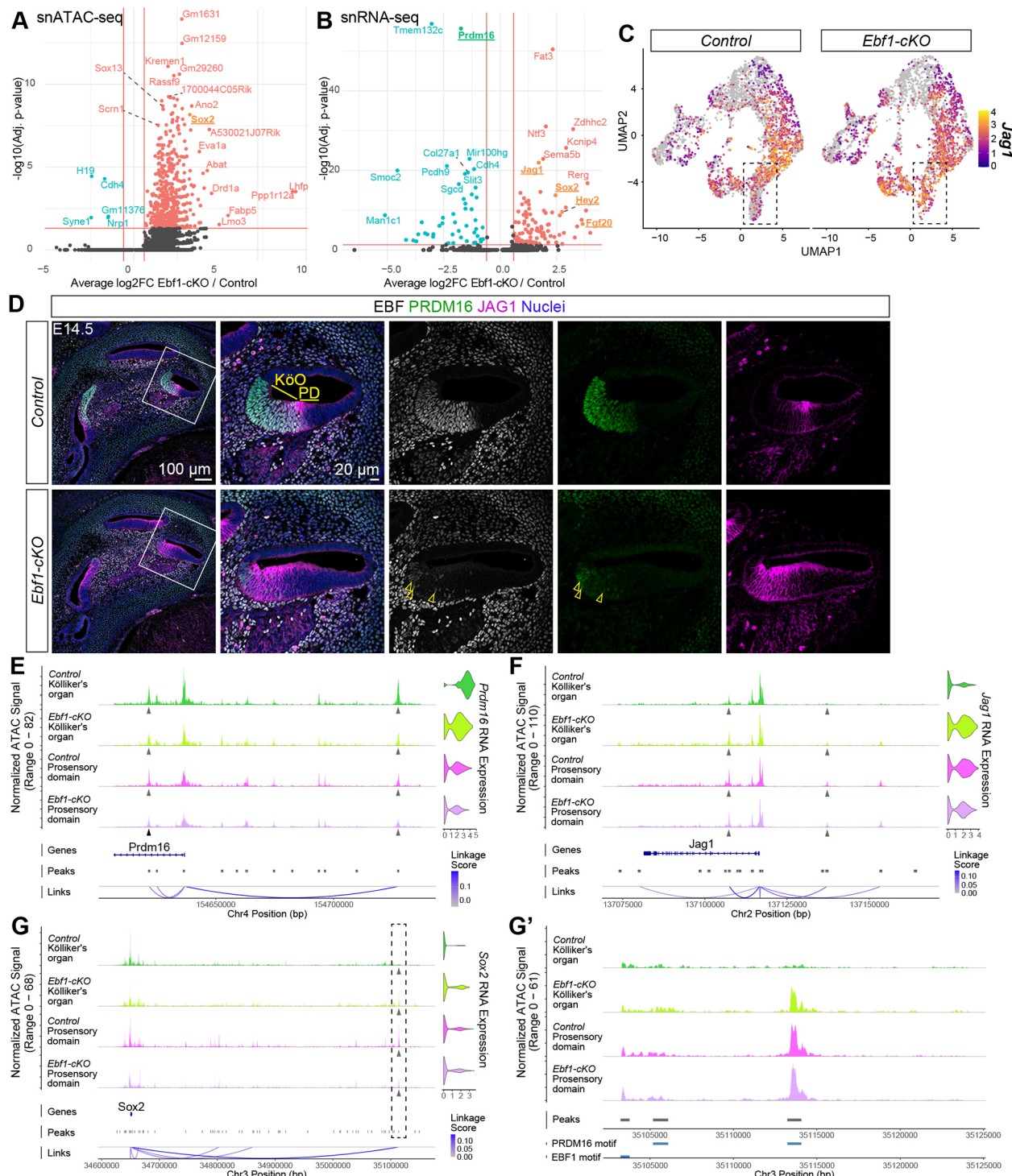

**Fig. 4. E14.5 multiomic analysis identifies potential EBF1 transcriptional targets involved in preventing the recruitment of Kölliker's organ cells into the prosensory domain**. (A,B) snATAC-seq volcano plot indicates differentially accessible peaks (A) and snRNA-seq volcano plot shows differentially expressed genes (B) in Kölliker's organ cells of E14.5 *Ebf1*-cKOs relative to control littermates. (C) E14.5 snRNA-seq feature plots split by genotype highlight *Jag1* upregulation in the Kölliker's organ. Dashed boxes indicate regions within the feature plots that contain nuclei assigned to the Kölliker's organ. (D) Confocal images of E14.5 mid-modiolar sections from control and *Ebf1*-cKO littermates capture an EBF1-dependent medial expansion of the JAG1 expression domain beyond the boundary between the Kölliker's organ (KöO) and prosensory domain (PD). White boxes (left) indicate regions included in the zoomed-in images (right). (E) Multiomic analysis of *Prdm16* gene expression and chromatin accessibility in E14.5 control and *Ebf1*-cKO cells from the Kölliker's organ and prosensory domain. Top left: coverage plots generated from snATAC-seq data. Gray arrowheads indicate ATAC peaks that are strongly correlated with *Prdm16* RNA expression and contain the EBF1-binding motif. Top right: *Prdm16* expression violin plots generated from snRNA-seq data. Bottom: Peak-to-gene analysis indicates strength of correlation between ATAC peak accessibility and *Prdm16* RNA expression. (F,G) Similar analyses are shown for ATAC peaks containing both EBF1 and PRDM16 binding motifs that are correlated with *Jag1* expression (F) and an ATAC peak containing the PRDM16 binding motif that is correlated with *Sox2* expression (G). (G′) Zoomed-in view of the ATAC peak ~500 kb upstream of the *Sox2* locus (dashed box in G) that contains the PRDM16 binding motif but not the EBF1-binding motif (bottom track).

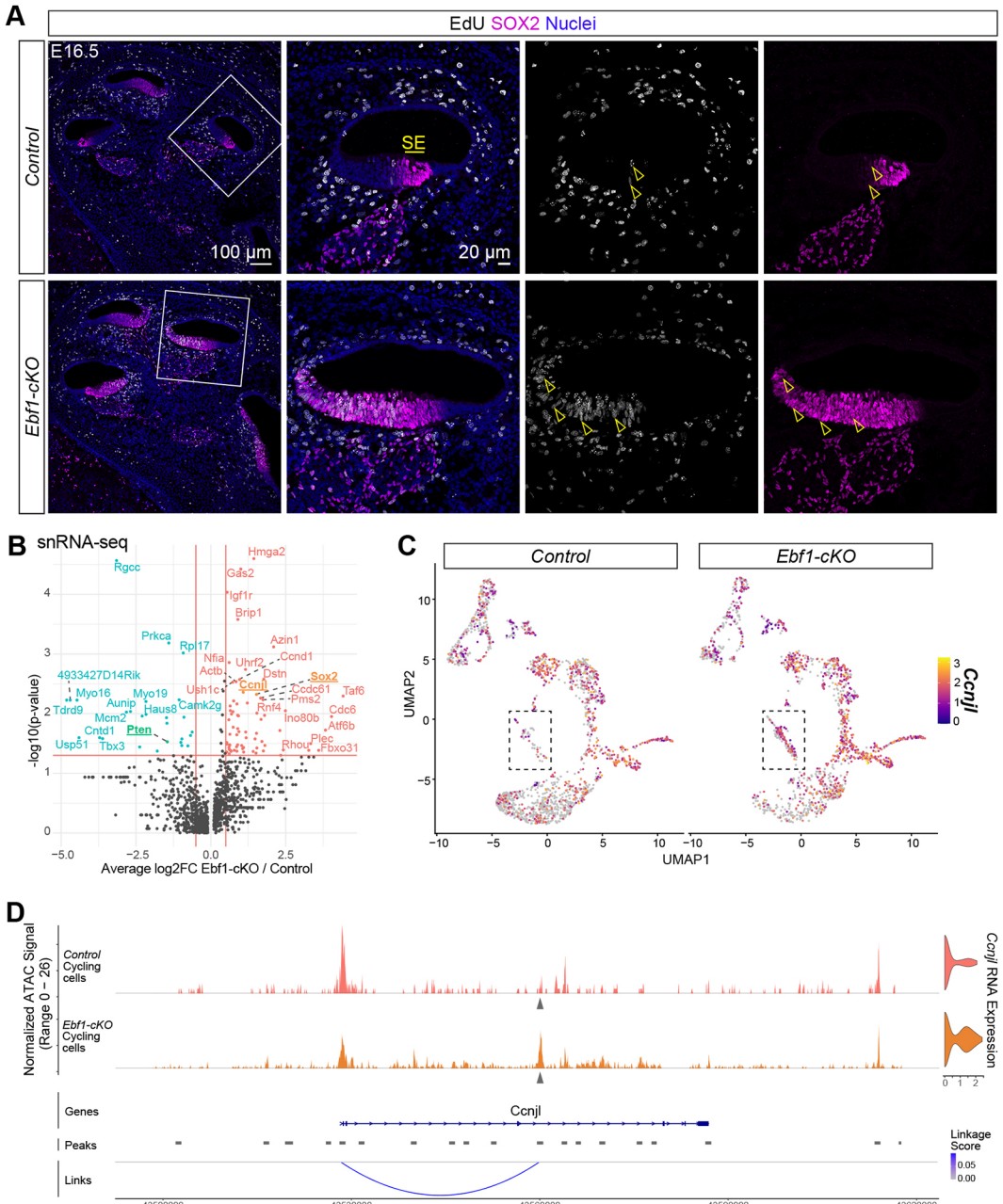

**Fig. 5. E16.5 multiomic analysis identifies an EBF1 transcriptional target potentially important in promoting prolonged prosensory cell proliferation**. (A) Confocal images of E16.5 mid-modiolar sections from control and *Ebf1*-cKO littermates treated with EdU 2 h before tissue collection. White boxes (left) indicate regions included in the zoomed-in images (right). (B) E16.5 snRNA-seq volcano plot shows differentially expressed cycling genes in the cycling cells of *Ebf1*-cKOs relative to control littermates. (C) Control and *Ebf1*-cKO snRNA-seq feature plots split by genotype highlight *Ccnjl* upregulation in *Ebf1*-cKO cycling cells. Dashed boxes indicate regions within the feature plots that contain nuclei assigned to the cycling cells. (D) Multiomic analysis of E16.5 control and *Ebf1*-cKO cycling cells. Top left: *Ccnjl* coverage plot generated from snATAC-seq data. Top right: *Ccnjl* expression violin plots generated from snRNA-seq data. Bottom: Peak-to-gene analysis shows strength of correlation between ATAC peak accessibility and RNA expression. Gray arrowheads indicate the ATAC peak that is strongly correlated with *Prdm16* RNA expression and contains both EBF1 and SOX2 binding motifs.

*Ebf1*-cKO relative to control cycling cells and identified *Pten*, *Sox2* and *Ccnjl* as possible targets of EBF1 (Fig. 5B; Table S2). PTEN is a well-established tumor suppressor and is downregulated in *Ebf1*-cKO cycling cells (Li et al., 1997; Steck et al., 1997). *Sox2*, previously noted for its role in inner ear sensory development, is also linked to stemness and tumor progression and is upregulated in these cells (reviewed by Boiani and Schöler, 2005; Boyer et al., 2005). *Ccnjl* encodes a cyclin-like protein associated with several cancers (Gu et al., 2024; Han et al., 2025; Li et al., 2021, 2025; Qin et al., 2024;

Viet et al., 2021; Wang et al., 2025) and is upregulated in *Ebf1*-cKO cycling cells (Fig. 5C). Of these candidates, *Ccnjl* emerged as the strongest for a direct EBF1 target involved in cell cycle exit. Neither *Pten* nor *Sox2* possess EBF1-binding motifs in open chromatin regions within or around their gene loci that are linked to their RNA expression, ruling them out as direct EBF1 targets. *Ccnjl*, by contrast, possesses an open chromatin region with an EBF1 motif that is more accessible in *Ebf1*-cKO than control cycling cells and strongly correlated with *Ccnjl* RNA expression (Fig. 5D). This same region

also possesses a SOX2 motif, suggesting that SOX2 directly promotes *Ccnjl* expression. While the effects of EBF1 on cell cycle exit may be mostly indirect, these results raise the possibility that EBF1 limits proliferation in the cochlear epithelium by repressing *Ccnjl* expression both directly and indirectly, by downregulating *Sox2*.

## DISCUSSION
### Proposed mechanism for the role of EBF1 in restricting cochlear sensory development

We highlighted the importance of EBF1 in restricting sensory development within the cochlea in our previous study (Powers et al., 2024). In this study, we use multiome sequencing to investigate the mechanism by which EBF1 regulates two aspects of prosensory domain establishment: (1) the positioning of the medial boundary of the prosensory domain and (2) the proliferative capacity of the sensory progenitor pool. We generated snRNA-seq and snATAC-seq libraries from the same nuclei isolated from cochlear ducts collected at two key developmental time points. This allowed us to screen differentially expressed genes from cell-type-matched populations in the *Ebf1*-cKO and control datasets for differentially accessible chromatin regions containing binding motifs associated with EBF1 or its direct targets. Analysis of E14.5 Kölliker's organ cells revealed that EBF1 likely prevents these cells from taking on a prosensory fate by directly regulating *Prdm16* and *Jag1* expression and indirectly regulating *Sox2* expression (Fig. 6A). A comparable regulatory relationship with *Prdm16* has been described for brown adipose differentiation, during which EBF2 directly promotes *Prdm16* expression (Rajakumari et al., 2013). PRDM16 is also known to act as a coregulator (Seale et al., 2007), and the transcription factor is thought to interact with EBF2 to regulate gene expression during brown adipogenesis (reviewed by Mao et al., 2024). We similarly see evidence that EBF1 and PRDM16 co-repress *Jag1* expression in the developing cochlea. Our multiomic analyses additionally provide evidence that PRDM16 directly represses *Sox2* expression. *Sox2*, however, is not uniformly downregulated in the PRDM16+ Kölliker's organ and maintains weak expression in Kölliker's organ cells closest to the E14.5 prosensory domain and E16.5 developing SE before becoming restricted to differentiated SCs (Figs 2A and 3A; Dabdoub et al., 2008). This pattern suggests that

other signals counteract PRDM16 activity and prevent complete repression of *Sox2* expression in this region. Our model of the EBF1 direct and indirect interactions with these transcriptional targets is supported by the finding that *Jag1* and *Sox2* are upregulated in *Prdm16* null cochleae (Ebeid et al., 2022). Multiome analysis of a later developmental stage, E16.5, revealed that EBF1 likely promotes cell cycle exit by directly repressing *Ccnjl* expression and indirectly repressing it through downregulation of *Sox2* (Fig. 6B). In zebrafish, SOX transcription factors show significant co-occurrence with EBF1 motifs in regeneration-responsive elements in SCs (Jimenez et al., 2022), suggesting potential regulatory interplay. On the basis of our results, we propose that EBF1 regulates sensory development in the cochlea through direct transcriptional control of key regulators of prosensory identity and proliferation. Delayed cell cycle exit combined with an increase in the size of the prosensory domain allows for the over twofold increase in HCs and SCs seen in *Ebf1*-cKOs (Kagoshima et al., 2024; Powers et al., 2024).

In another developmental context, B cell lineage specification, EBF1 is known to recruit chromatin remodeling complexes that promote chromatin accessibility (reviewed by Ramírez et al., 2010; Wang et al., 2020; Zolotarev et al., 2022). In the cochlea, however, *Ebf1* excision leads to an overall increase in chromatin accessibility in Kölliker's organ cells at E14.5 (Fig. 4A). To gain mechanistic insight into this change, we searched for motif enrichment within the differentially accessible chromatin regions and found that the SOX2 motif was the most significantly enriched of the motifs belonging to transcription factors expressed in the snRNA-seq data. SOX2 is known to act as an activator during hippocampal neurogenesis and cochlear development (Ahmed et al., 2012; Amador-Arjona et al., 2015; Kempfle et al., 2016), and the upregulation of *Sox2* in *Ebf1*-cKO Kölliker's organ cells likely drives the overall increase in their chromatin accessibility. Interestingly, while the changes in gene expression in *Ebf1*-cKO relative to control Kölliker's organ are similar at E14.5 and E16.5 (219 upregulated and 105 downregulated genes in E16.5 Kölliker's organ; Fig. S7B), open chromatin accessibility at E16.5 is less skewed towards an overall increase and not linked to a specific transcription factor or pathway (183 more accessible and 113 less accessible ATAC peaks in E16.5 Kölliker's organ; Fig. S7A), suggesting that this SOX2-driven change is specific to E14.5.

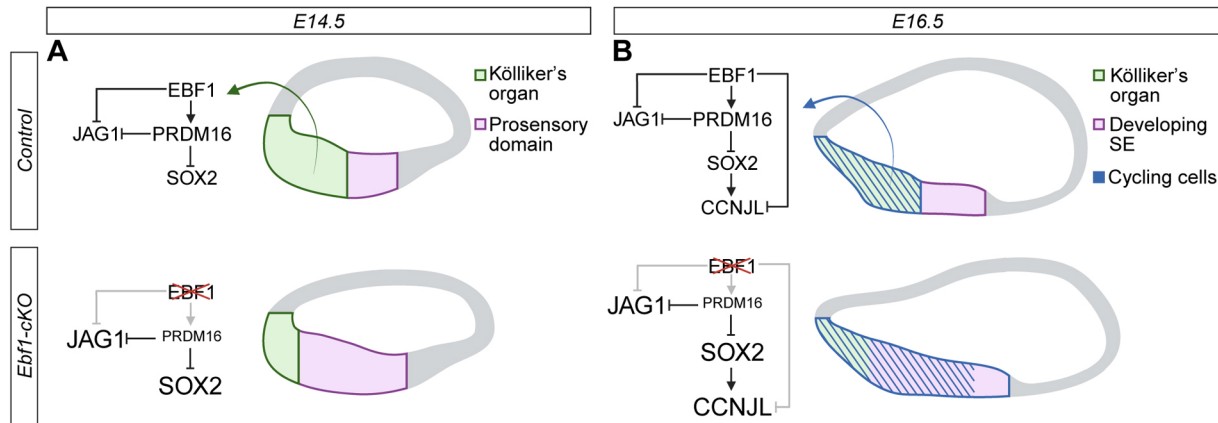

**Fig. 6. EBF1 restricts prosensory domain establishment by both positioning the medial boundary of the prosensory domain and restricting prosensory cell proliferation.** (A,B) Cross-sectional diagrams of control and *Ebf1*-cKO cochleae. Created in BioRender by Powers, K, 2026. https://BioRender.com/ij02m05. This figure was sublicensed under CC-BY 4.0 terms. (A) At E14.5, EBF1 appears to directly promote *Prdm16* expression and co-represses *Jag1* expression with PRDM16 and, in turn, PRDM16 directly represses *Sox2* expression. Consequently, loss of EBF1 leads to *Prdm16* downregulation in addition to *Jag1* and *Sox2* upregulation in the Kölliker's organ, driving these cells towards a more prosensory fate. (B) At E16.5, in addition to promoting *Prdm16* expression and repressing *Jag1* and *Sox2* expression, EBF1 seems to be needed to repress *Ccnjl* expression in cycling cells. Loss of EBF1 removes direct repression of *Ccnjl*, while concurrent *Sox2* upregulation promotes further *Ccnjl* expression.

## EBF1 interacts with the Notch pathway during cochlear patterning

EBF1 is known to repress gene expression in a dose-dependent manner (Lukin et al., 2011), and Notch signaling has been shown to reduce EBF activity (Smith et al., 2005). Thus, EBF1 expression may need to reach a certain threshold relative to Notch signaling levels to downregulate *Jag1*. At E12.5, EBF1 and JAG1 expression domains overlap, with EBF1 restricted to the medial edge (Kagoshima et al., 2024; Powers et al., 2024) and JAG1 broadly expressed throughout the cochlear floor (Ohyama et al., 2010). At E14.5, JAG1 expression shifts from Kölliker's organ to the prosensory domain (Basch et al., 2016; Murata et al., 2006), while EBF1 is strongly expressed in the Kölliker's organ and weakly in the prosensory domain (Fig. 4D; Kagoshima et al., 2024; Powers et al., 2024). JAG1 becomes increasingly restricted to differentiated SCs starting at around E16.5 (Fig. S7C,D; Murata et al., 2009). At the same stage, EBF1 is only weakly expressed in the developing SE but strongly expressed in the Kölliker's organ, especially in the region of the organ closest to the developing SE (Fig. S7D), and later persists at low levels in adult SCs (Powers et al., 2024). The interplay between EBF1 and JAG1 expression may allow EBF1 to reach the appropriate threshold, relative to Notch activity levels (Murata et al., 2006), needed to downregulate *Jag1* in the Kölliker's organ at around E14.5. Relatively low EBF1 levels in the developing SE at E16.5, together with high Notch activity associated with sensory patterning through lateral inhibition (Murata et al., 2006), likely permit JAG1 expression to persist in this portion of the cochlear floor, whereas high EBF1 expression at the lateral edge of the Kölliker's organ may be necessary to maintain the medial boundary of the developing SE.

Just as BMP signaling is essential for establishing the lateral boundary of the prosensory domain (Ohyama et al., 2010), Notch signaling has been proposed to be crucial for establishing the medial boundary (Basch et al., 2016; Maunsell et al., 2023). As JAG1 expression shifts from the Kölliker's organ to the differentiating prosensory domain, its expression becomes localized to a narrow band at the medial boundary of the prosensory domain (Fig. 4D). Notch signaling at this boundary is essential for establishing a single row of iHCs (Basch et al., 2016). The differentiated nascent iHCs send Notch signals to neighboring cells in the Kölliker's organ, preventing them from becoming iHCs or inner phalangeal cells. A modest reduction in Notch signaling has been reported to result in supernumerary iHCs and inner phalangeal cells. Double mutants for *Lfng* and *Mfng*, the genes encoding glycosyltransferases that strengthen Notch receptor affinity for DLL1 over JAG1, possess duplicates of iHCs and inner phalangeal cells (Basch et al., 2016). A similar phenotype is seen in mice with mutations in *Lrrn1*, which encodes a single-pass membrane protein that facilitates Notch signaling along the medial boundary of the prosensory domain (Maunsell et al., 2023). By contrast, a more pronounced reduction in Notch signaling impairs lateral inhibition, leading to the formation of supernumerary HCs at the expense of SCs (Kiernan et al., 2005a, 2006). Neonatal *Ebf1*-cKOs possess multiple rows of iHCs and these sensory cells are accompanied by supernumerary inner phalangeal cells (Powers et al., 2024). The expanded JAG1 expression domain in E14.5 *Ebf1*-cKOs is consistent with an increase in iHC and inner phalangeal precursors at the medial boundary of the prosensory domain due to aberrant Notch signaling.

## Understanding ectopic sensory patch formation in *Ebf1*-cKOs

Loss of EBF1 leads to the formation of ectopic sensory patches (Powers et al., 2024), likely derived from the partial conversion of the Kölliker's organ to SE. Other manipulations, particularly in the Notch pathway, have shown similar phenotypes. For example, ectopic expression of ATOH1, a transcription factor required for HC differentiation in the inner ear, is sufficient to generate HCs in the Kölliker's organ (Woods et al., 2004; Zheng and Gao, 2000). Similarly, Notch signaling promotes and can directly induce sensory formation in non-SE within the inner ear (Driver et al., 2008; Hartman et al., 2010; Neves et al., 2011; Pan et al., 2010). The upregulation of *Jag1* in *Ebf1*-cKO Kölliker's organ cells likely leads to increased Notch activity and, ultimately, to the ectopic sensory patches in the Kölliker's organ by neonatal stages (Powers et al., 2024). A similar ectopic patch phenotype is seen in neonatal *Prdm16* null cochlea, which also show an upregulation of *Jag1* relative to controls (Ebeid et al., 2022; Zhang et al., 2025). Interestingly, the *Ebf1*-cKO PRDM16 expression domain is less diminished at E16.5 (Fig. 3A and Fig. S7) than at E14.5 (Figs 2A and 4D). EBF3 and EBF4, which are expressed in the Kölliker's organ at moderate levels during cochlear development (Powers et al., 2024), may limit the extent to which Kölliker's organ cells adopt a prosensory fate in the absence of EBF1. In keeping with this hypothesis, *Ebf3* is upregulated in Kölliker's organ cells of E14.5 *Ebf1*-cKOs compared with controls (Table S1). The conversion of the Kölliker's organ to SE is likely patchy, rather than a completely confluent transformation, because of compensatory mechanisms.

## EBF1 interactions with regulators of cell cycle

Loss of EBF1 leads to delayed cell cycle exit in the developing cochlea (Fig. 5A; Powers et al., 2024). Prolonged prosensory proliferation often leads to the generation of supernumerary HCs and SCs (Chen and Segil, 1999; Jacques et al., 2012; Kiernan et al., 2005a; Löwenheim et al., 1999; Tateya et al., 2011) and contributes to the over twofold increases in sensory cells seen in *Ebf1*-cKOs by neonatal stages (Kagoshima et al., 2024; Powers et al., 2024). EBF1 is known to regulate cell cycle exit during neurogenesis and B cell lineage specification, and emerging evidence points to EBF, particularly EBF1 and EBF3, involvement in tumor suppression (reviewed by Liao, 2009). During neurogenesis, EBFs couple differentiation and migration to cell cycle exit. Ectopic expression of *Ebf1* in neuroepithelial progenitors leads to their premature exit from the cell cycle (Garcia-Dominguez et al., 2003). In early-stage B cells, EBF1 regulates expression of cell cycle regulator genes, including *E2f2*, *E2f8*, *Cdk2*, *Ccnd2*, *Ccnd3* and *Cdc6* (Györy et al., 2012). Though we do not see evidence that EBF1 directly regulates these targets in our E16.5 multiome datasets, we do see evidence that EBF1 directly downregulates *Ccnj1* in cycling cells.

## Study limitations

We used multiome sequencing to simultaneously profile gene expression and chromatin accessibility in the same nuclei isolated from the cochleae of *Ebf1*-cKO and control littermates. We used this information to identify cell types, compare differential gene expression and chromatin accessibility in cell-type-matched populations, and infer transcriptional regulation by EBF1 and its downstream targets. Rather than focusing on EBF1 targets alone, this approach provides a global view of how EBF1 fits into a regulatory network while also minimizing the confounding noise that comes from analyzing an entire tissue with cell types as diverse as those of the mammalian cochlea. A limitation of our study is that we have not confirmed direct binding of EBF1 to the proposed transcriptional targets. In future studies, chromatin immunoprecipitation sequencing (ChIP-seq; Solomon and Varshavsky, 1985), cleavage under targets and release using nuclease (CUT&RUN; Skene and Henikoff, 2017) or cleavage under targets and tagmentation (CUT&Tag; Kaya-Okur et al., 2019) could be used to accomplish this.

## Conclusions

Our findings place EBF1 as an integral regulator within the gene regulator networks controlling cochlear development. By E14.5, EBF1 likely activates *Prdm16* and represses *Jag1* and *Sox2* to prevent Kölliker's organ cells from taking on a prosensory fate and help position the medial boundary of the prosensory domain. By E16.5, EBF1 may promote cell cycle exit in the cochlear epithelium by downregulating *Ccnj1* directly and indirectly, through its downregulation of *Sox2*. Together, the medial shift of the prosensory domain and delayed cell cycle exit drive the pronounced sensory expansion in *Ebf1*-cKOs. EBF1 plays an essential role in restricting sensory establishment within the developing cochlea.

## MATERIALS AND METHODS

### Animal care and strains

Mice were housed at the University of Washington Department of Comparative Medicine. All experiments were approved by the Institutional Animal Care and Use Committee (IACUC) of the University of Washington and performed in accordance with the standards outlined by the National Institutes of Health (NIH). Mice were euthanized in accordance with IACUC approved procedures and in line with NIH policies.

*Ebf1*$^{fl/fl}$ mice were obtained from The Jackson Laboratory on a C57BL/6 background (strain #028104). The *Slc26a9*$^{P2A-Cre}$ mice were obtained from a colleague on a C57BL/6 background (Urness et al., 2020) and are available from MMRRC, UC Davis, CA, USA (RRID:MMRRC_067348-MU). To generate *Slc26a9*-conditional *Ebf1* knockout mice, *Slc26a9*$^{P2A-Cre}$ males were crossed with *Ebf1*$^{fl/fl}$ females to generate *Slc26a9*$^{P2A-Cre}$ *Ebf1*$^{fl/+}$ progeny. Male *Slc26a9*$^{P2A-Cre}$ *Ebf1*$^{fl/+}$ mice identified by genotyping (see below) were then crossed with female *Ebf1*$^{fl/fl}$ mice to generate litters containing *Slc26a9*$^{P2A-Cre}$ *Ebf1*$^{fl/fl}$ mice. *Slc26a9*$^{P2A-Cre}$ *Ebf1*$^{fl/fl}$ male mice identified by genotyping were bred with *Ebf1*$^{fl/fl}$ female mice to generate *Slc26a9*$^{P2A-Cre}$ *Ebf1*$^{fl/fl}$ (*Ebf1*-cKO) and *Ebf1*$^{fl/fl}$ (control) littermates for phenotypic and multiomic analyses. Embryonic stages were verified by Theiler's criteria.

### Genotyping

Tail tips or ear punches were collected for genotyping. Mice for multiome sequencing were screened for *Slc26a9*$^{P2A-Cre}$ using the following qPCR primer pair: GGTGCAAGCTGAACAACAGG and CAGGTGCTGTTG-GATGGTCT. All other mice were screened for *Slc26a9*$^{P2A-Cre}$ using the following PCR primer trio: GGAGGAACACAGTTCACAGT, GTGT-CTGGTGTGGCTGATGACC and ATGGGTTCACCAGAGTCTCATC. All mice were genotyped to identify homozygosity for the floxed *Ebf1* allele (*Ebf1*$^{fl/fl}$) using two sets of qPCR primer pairs: (1) TGTGGCAACC-GAAATGAG and CCTGTGAGCGACACAAAGC in addition to (2) ACGACTTCTTCAAGTCCGCC and TCTTGTAGTTGCCGTCGTCC. The first primer pair was designed to identify the presence of the wild-type *Ebf1* allele (Vilagos et al., 2012). The second primer pair was used to reveal the presence of the *GFP* associated with the fusion protein encoded in the floxed *Ebf1* allele (Vilagos et al., 2012). Mice were identified as *Ebf1*$^{fl/fl}$ if they tested negative for the wild-type *Ebf1* allele and positive for *GFP*.

### Immunostaining and EdU labeling

E14.5 and E16.5 heads were fixed overnight in 4% paraformaldehyde at 4°C. Following fixation, heads were washed in PBS and incubated in a sucrose series consisting of 5%, 10%, 15%, 20%, 25% and 30% sucrose washes and a final wash in a 1:1 solution of OCT (Tissue-Tek, 4583) and 30% sucrose. Heads were then incubated in OCT for 3 h at room temperature before being embedded in OCT and frozen for sectioning.

The 12 μm tissue sections were washed with 0.1% Triton X-100 in PBS before being blocked for 1 h at room temperature with a blocking solution consisting of 0.5% Triton X-100 and 10% normal donkey serum in PBS. Primary antibodies – mouse anti-EBF (C-8) Alexa Fluor-647 (Santa Cruz Biotechnology, sc137065-af647, 1:200), rabbit anti-JAG1 (Cell Signaling Technology, 2620, 1:125), sheep anti-PRDM16 (R&D Systems, af6295, 1:200), goat anti-SOX2 (R&D Systems, af2018, 1:200) and rabbit anti-SOX2 (Abcam, ab97959, 1:200) – were diluted in blocking solution and applied to the tissue sections overnight at room temperature. Sections

were then washed with 0.1% Triton X-100 in PBS and labeled with Alexa Fluor-labeled secondary antibodies – donkey anti-sheep Alexa Fluor 488 (Invitrogen, A11015, 1:400), donkey anti-goat Alexa Fluor 488 (Invitrogen, A11055, 1:400) and donkey anti-rabbit Alexa Fluor 568 (Invitrogen, A10042, 1:400) – and Hoechst 33342 (Invitrogen, H3570, 1:10,000) for 1 h at room temperature.

To detect proliferating cells at E16.5, 5-ethynyl-2′-deoxyuridine (EdU; 50 mg/kg; Invitrogen, A10044) reconstituted in sterile PBS was administered to pregnant dams via intraperitoneal injection, 2 h before tissue collection. Click-iT EdU Cell Proliferation Kit for Imaging with Alexa Fluor 647 dye (Invitrogen, C10340) was used to detect incorporated EdU in E16.5 cochleae using the manufacturer's protocol. All tissue sections were imaged on a Zeiss LSM 880 confocal microscope.

### Single nucleus multiome library preparation and barcoding

After collecting E14.5 or E16.5 embryos from *Ebf1*$^{fl/fl}$ females timed-mated with *Slc26a9*$^{P2A-Cre}$ *Ebf1*$^{fl/fl}$ males, tail snips were gathered for *Slc26a9*$^{P2A-Cre}$ qPCR genotyping, and the temporal bones were removed. The cochlear ducts were then dissected in 1% 1 M HEPES diluted in HBSS and pooled according to genotype in cryogenic tubes. The samples were centrifuged at 300 **g** for 6 min at 4°C, after which the saline solution was removed using an insulin syringe and the cochlear ducts were snap frozen and stored in liquid nitrogen. Once E14.5 and E16.5 samples with at least six *Slc26a9*$^{P2A-Cre}$ *Ebf1*$^{fl/fl}$ and six *Ebf1*$^{fl/fl}$ cochlear ducts had been obtained, the samples were thawed on ice and the nuclei were isolated using a modified version of the protocol for the Chromium Nuclei Isolation Kit (10x Genomics, 1000493) that omitted steps involving vortexing or use of a pestle or column. Briefly, samples were incubated in 500 μl lysis buffer for 1 min on ice, then triturated until fully dissociated (∼5-6 min). Next the nuclei suspensions were filtered through cell strainer caps (FALCON, 352235) into pre-chilled collection tubes and spun down at 500 **g** for 3 min at 4°C. The nuclei pellets were then washed with 500 μl of a 1:1 solution of wash buffer and resuspension buffer, centrifuged at 500 **g** for 10 min at 4°C, and resuspended in 500 μl resuspension buffer. Lysis, wash and resuspension buffers were made using the recipes in the Chromium Nuclei Isolation Kit user guide.

Following nuclei isolation and purification, samples were centrifuged at 300 **g** for 5 min at 4°C and resuspended in HBSS to reach a concentration of 5000 cells/μl. Library construction was carried out with a 9000 targeted nuclei recovery using the Chromium Next Cell Multiome ATAC+Gene Expression Reagent Bundle (10x Genomics, 10000285) following the manufacturer's protocol. Briefly, nuclei were encapsulated in a gel matrix and uniquely barcoded using the 10x Chromium Controller (Chip J, 10x Genomics, 1000230). Libraries were then sequenced using Illumina NovaSeq.

### Single nucleus multiome sequencing data preparation and analysis

CellRanger arc count (2.0) was used to align sequencing reads to the mm10 genome and define ATAC peaks. The aligned data was loaded into R and further analyzed using the Signac (v1.14.0; Stuart et al., 2021) and Seurat (v5.2.1; Hao et al., 2024) packages. *Ebf1*-cKO and control samples from the same time point were combined and analyzed together. To ensure consistent peak definitions, overlapping or adjacent genomic intervals between the two conditions were merged into a single peak, and the peak expression matrix was calculated using the new merged peak set in Signac using 'FeatureMatrix' and stored in the ATAC assay.

Next, the gene expression matrix was stored in the RNA assay and processed using the Seurat pipeline. Reads were log-normalized using 'NormalizeData', 2000 variable features were identified using 'FindVariableFeatures', and data integration using CCA was performed to correct for batch effects between the two conditions. The batch-corrected expression of each gene was scaled and used to calculate principal components (PCs) using RunPCA(). The top 20 PCs were used to identify neighbors, find clusters and generate a UMAP. Canonical markers were used to label clusters epithelium (*Epcam*), immune cells (*C1qa*), spiral ganglion (*Nefm*), Schwann cells (*Fabp7*), mesenchyme (*Pou3f4*) and endothelium (*Pecam1*) at E14.5 and E16.5 (Figs S1 and S4) in addition to melanocytes (*Pmel*) at E16.5 (Fig. S4). As we were primarily interested in the SE, epithelial cells were separated, and the clustering procedure was repeated on this subset of cells. Canonical markers were used to label the epithelial

clusters. E14.5 and E16.5 epithelial clusters were annotated as Kölliker's organ (*Tecta*, *Fgf10* and *Prdm16*), prosensory domain or developing SE (*Tecta*, *Hey2* and *Sox2*), early HCs or HCs (*Tecta*, *Sox2*, *Atoh1*, *Pou4f3* and *Gfi1*), outer sulcus (*Bmp4* and *Fst*), cycling cells (*Top2a*), stria vascularis (*Oc90*) and Reissner's membrane (*Oc90* and *Otx2*) (Figs S2 and S5). E16.5 epithelial clusters were also labeled as iHCs (*Fgf8*), oHCs, and early lateral SCs (*Hes5*, *Heyl* and *Ngfr*) (Fig. S5).

Differential expression tests were performed between conditions using a Wilcoxon ranked-sum test implemented in Seurat's 'FindMarkers' command with default parameters. This was applied to find differences in gene expression using the RNA assay and peak accessibility using the ATAC assay.

Cell-cell communication analysis was performed using CellChat (v2.1.2; Jin et al., 2021). Analysis was performed separately for the *Ebf1*-cKO and control conditions, and non-protein signaling interactions were excluded. The average gene expression by cell type was calculated using the trimean and used to infer cell-cell communication probabilities.

## Peak-gene and transcription factor-gene linkage
After removing peaks associated with contigs, peaks were analyzed for transcription factor binding sites using the Signac 'AddMotifs' function. The Jaspar 2020 database was used for transcription factor motifs. Signac's 'RunChromvar' function was used to estimate transcription factor accessibility within individual cells. To link peak accessibility to gene expression, we used 'LinkPeaks' and searched for peaks within 500 kb of the transcription start site of each gene. rGREAT (v2.4.0; Gu and Hübschmann, 2023) was used to perform gene ontology analysis of the differentially accessible peaks.

## Acknowledgements
We thank Dr Matsya Ruppari Thulasiram from Dr Alain Dabdoub's lab for her help with optimizing the protocols used to snap freeze embryonic cochlear ducts in liquid nitrogen and to isolate their nuclei. We also thank the Northwest Genomics Center for their help sequencing our multiome libraries. Lastly, we thank Dr Thomas Reh for comments on the manuscript and Reh lab members for insightful discussions regarding this study.

## Competing interests
The authors declare no competing or financial interests.

## Author contributions
Conceptualization: O.B.-M.; Data curation: K.G.P., J.H.; Formal analysis: K.G.P., J.H., O.B.-M.; Funding acquisition: K.G.P., O.B.-M.; Investigation: K.G.P., J.H., O.B.-M.; Methodology: K.G.P., J.H., J.W.; Project administration: O.B.-M.; Software: J.H.; Supervision: O.B.-M.; Visualization: J.H.; Writing – original draft: K.G.P., J.H., J.W., O.B.-M.; Writing – review & editing: K.G.P., J.H., J.W., O.B.-M.

## Funding
This study received funds from the National Institute on Deafness and Other Communication Disorders (R01DC017126 to O.B.-M.) and the University of Washington Institute for Stem Cell and Regenerative Medicine Fellows Program (to K.G.P.). Open Access funding provided by the University of Washington. Deposited in PMC for immediate release.

## Data and resource availability
All sequencing data for RNA and ATAC analysis have been deposited in the GEO database under accession number GSE310246. Analysis scripts have been uploaded on Github (https://github.com/joshhahn28/EBF1-cochlea-multiome). All other relevant data and details of resources can be found within the article and its supplementary information.

## Peer review history
The peer review history is available online at https://journals.biologists.com/dev/lookup/doi/10.1242/dev.205207.reviewer-comments.pdf

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
