## [Peer Review File · Development (Cambridge, England)]

EBF1 regulates sensory establishment in the cochlea by positioning the medial boundary of the prosensory domain and restricting proliferation of the sensory progenitor population

Kathryn G. Powers, Joshua Hahn, Juliette Wohlschlegel and Olivia Bermingham-McDonogh
DOI: 10.1242/dev.205207

Editor: James Briscoe

Review timeline

Original submission: 2 September 2025
Editorial decision: 9 October 2025
First revision received: 17 December 2025
Accepted: 4 January 2026

Original submission

First decision letter

MS ID#: dev.205207

MS TITLE: EBF1 regulates sensory establishment in the cochlea by positioning the medial boundary of the prosensory domain and restricting proliferation of the sensory progenitor population

AUTHORS: Kathryn G. Powers, Joshua Hahn, Juliette Wohlschlegel and Olivia Bermingham-McDonogh

Dear Dr Bermingham-McDonogh,

I have now received all the referees' reports on the above manuscript, and have reached a decision. The referees' comments are appended below, or you can access them online: please go to:

As you will see, the referees express considerable interest in your work, but have some significant criticisms and recommend a substantial revision of your manuscript before we can consider publication. Most importantly, the referees would like to see functional validation for your proposed regulatory mechanisms, as the current evidence is correlative. Addition of data or other evidence demonstrating that Prdm16 and Ccnjl are bona fide direct targets of Ebf1 and/or experiments to establish the proposed functions of the hypothesised targets would greatly strengthen the study. Approaches such as ChIP, overexpression, or knockdown studies in cochlear explants, as suggested by the reviewers would address this issue. Alternatively, if such experiments are not possible, please make clear in your discussion that these proposals are hypotheses that will require functional validation in future work. Additionally, please respond to the other points raised by the referees.

If you are able to revise the manuscript along the lines suggested, which may involve further experiments, I will be happy to receive a revised version of the manuscript. Your revised paper will be re-reviewed by one or more of the original referees, and acceptance of your manuscript will depend on your addressing satisfactorily the reviewers' major concerns. Please also note that Development will normally permit only one round of major revision. If it would be helpful, you are welcome to contact us to discuss your revision in greater detail. Please send us a point-by-point response indicating your plans for addressing the referees' comments, and we will look over this and provide further guidance.

Please attend to all of the reviewers' comments and ensure that you upload both a 'clean' version of your Word file, along with a highlighted version clearly showing where you have made changes in the revised manuscript. Please avoid using 'Tracked changes' in Word files as these are lost in PDF conversion. I should be grateful if you would also provide a point-by-point response detailing how you have dealt with the points raised by the reviewers in the 'Response to Reviewers' box. If you do not agree with any of their criticisms or suggestions please explain clearly why this is so.

Reviewer 1

Advance summary and potential significance to field

This manuscript is a follow up to a recent paper from the Birmingham McDonogh lab on the role of the EBF1 transcription factor in the development of the cochlea. This tissue has great interest for developmental biologists due to its highly organized cellular structure where the numbers of its constituent cell types are tightly regulated. Moreover, the mammalian cochlea exhibits a remarkable decoupling of cell cycle exit from differentiation, whereby cells in the apical tip of the cochlea (that will ultimately detect low frequency sounds) exit the cell cycle first but then wait for four or five days before differentiating. In contrast, cells at the base (high frequency) region of the cochlea exit the cell cycle last, but almost immediately differentiate.

The authors previously showed that inactivation of the EBF1 gene led to a great over-production of hair cells and supporting cells in the cochlea, and that this over-production is due in part to a significant expansion of the Sox2+ prosensory domain from which these cells differentiate. The present study seeks a mechanistic explanation for these results by performing multiomic single cell analysis of the embryonic cochlear duct at two ages. They propose model in which EBF1 promotes expression of the transcription factor Prdm16 in the medial side of the cochlear duct. This gene in turn represses Sox2, a marker of the prosensory domain of the cochlear duct. In EBF1 knockouts, Prdm16 is largely lost, leading to an up-regulation of Sox2, an expansion of the prosensory domain and increased numbers of hair cells and supporting cells. They also provide evidence for EBF1 regulating a cell cycle component, CCNJL.

Comments for the author

The paper is well written, and the fluorescence images are of extremely high quality. Multiomic single cell analysis is tricky, and the authors should be congratulated for obtaining such clean and convincing results from relatively small amount of cochlear tissue. The data and bioinformatic analysis make a good case for the model proposed.

I have one minor concern with the model proposed by the authors and their interpretations. Sox2 is not simply restricted to the prosensory domain of the cochlear duct, but remains expressed in at least some cells of Kölliker's organ closer to the inner hair cell region. This domain of Sox2+ KO cells likely overlaps with Prdm16 expression. Sox2 is down-regulated from this region in a basal-apical gradient in the first few days after birth, but is certainly present in parts of KO at late embryonic ages. These cells do not normally give rise to hair cells and supporting cells and are not therefore "prosensory". How do the authors reconcile this with their model? Might EBF1 be repressing prosensory formation and hair cell and supporting cell differentiation independently of its repression of Sox2? If they examine their immunostaining closely, how many cells express both EBF1 and Sox2, and for how long between E14 and birth?

One more minor point - the acknowledge thank a colleague for helping with "optimizing the protocol used to snap freeze the embryonic cochlear ducts in liquid nitrogen". Since it would appear this protocol is critical to the success of the project, it would be helpful if the authors described this procedure in some detail.

Reviewer 2

Advance summary and potential significance to field

The manuscript by Powers et al. describes the impact of conditional Ebf1 (Ebf1 cKO) mutations on gene expression and chromatin accessibility during mouse cochlear development. This work follows a recent publication by the same group in *Development*, covering a similar topic. In the current study, single-cell multiome analysis at embryonic cochlear stages reveals changes in the expression of genes such as Prdm16, Sox2, Jag1, and Ccnjl in Ebf1 cKO mice, which may explain alterations in the size and position of the prosensory domain. Chromatin accessibility data, supported by motif analysis, suggest that some of these differentially expressed genes could be direct Ebf1 targets. The authors propose a gene regulatory module involved in establishing the medial boundary of the prosensory domain adjacent to Kolliker's organ. However, while the multiome approach suggests a potential mechanism, the study does not include experiments that directly test this regulatory model. Furthermore, most of the proposed regulatory relationships, except for Ccnjl, have previously been identified or predicted. Since this study represents only an incremental advance in our understanding of prosensory development, it is considered premature for publication in *Development*.

Comments for the author

The following is a list of major and minor concerns.

1. We highlighted the importance of EBF1 in restricting sensory development within the cochlea in our previous study (Powers et al., 2024). In this study, we use multiome sequencing to uncover the mechanism by which EBF1 regulates two aspects of prosensory domain establishment: (1) the positioning of the medial boundary of the prosensory domain and (2) the proliferative capacity of the sensory progenitor pool.

Multiome analysis may imply, but does not prove, mechanism. It was used to identify candidate targets of Ebf1 regulation however, no follow-up experiments were performed to rigorously address the authors' claims that Prdm1 and Ccnjl are direct targets of Ebf1. Consequently, the results are correlative and require functional studies to validate these claims. For example, is Prdm1 bound to the sites in question and is this recruitment required for target gene regulation?

2. Ccnjl, by contrast, possesses an open chromatin region with an EBF1 motif that is more accessible in Ebf1-cKO than control cycling cells and strongly correlated with Ccnjl RNA expression (Fig. 5D).

How does the weaker ATAC-seq peak at the Ccnjl promoter in Ebf1cko factor into the authors' model?

3. In between the rows of iHCs and oHCs are two rows of pillar cells (inner and outer) that differentiate with actin rich structures to form the tunnel of Corti (Fig. 1).

The aforementioned statement is inaccurate and should be amended as follows: Pillar cells are enriched with microtubule arrays that are maintained through actin crosslinks.

4. The sensory epithelium (SE) lies above the basilar membrane which vibrates in response to sound, allowing the stereocilia on the tops of the HCs to deflect and thereby causing neurotransmitter release from the iHCs.

The aforementioned statement lacks an important detail and should be amended as follows: The sensory epithelium (SE) lies above the basilar membrane, which vibrates in response to sound. This vibration causes deflection of the stereocilia on the tops of the HCs, initiating mechanotransduction and leading to neurotransmitter release from the IHCs.

Reviewer 3

Advance summary and potential significance to field

The study by Powers et al. offers new insights into the gene regulatory network that governs the position of the medial boundary between the auditory sensory epithelium and a group of transient epithelial cells termed Kölliker's cells. Defects in positioning this boundary alters auditory sensory development and results in hearing loss. In their recent *Development* paper, the authors demonstrated that early otic loss of the transcription factor EBF1 results in a significant medial expansion of the auditory sensory domain, accompanied by ectopic cell proliferation.

To uncover the underlying mechanisms, the authors conducted multiomic experiments using cochlear tissue from E14.5 and E16.5. By integrating single-nucleus RNA sequencing and ATAC sequencing data, they show that EBF1 prevents Kölliker's cells from adopting a sensory fate by activating the expression of *Prmd16* and by repressing the expression of *Jag1* and *Sox2*. Additionally, they present tentative evidence that EBF1 may promote cell cycle exit within the developing cochlear epithelium by repressing *Ccnjl* expression. Overall, this is a well-designed and executed study and I have only few suggestions for improvement.

Major Comment:

The authors indicate in the abstract and main text that EBF1 may promote cell cycle exit by repressing *Ccnjl* expression. However, no functional evidence is provided to support this. Unlike other identified *Ebf1* targets such as *Prdm16*, *Jag1*, and *Sox2*, which have established roles in auditory sensory development, the function of *Ccnjl* in cochlear development has yet to be established. To remedy this the authors could use an in vitro approach such as cochlear explants and overexpress *Ccnjl* (or overexpress/ knockdown other candidates) and evaluate its effect on cell proliferation.

Minor Comments:

- 1) The main text should specify the Cre strategy used for the conditional knockout of *Ebf1*.
- 2) Readme files for the supplemental tables should be included.

First revision

Author response to reviewers' comments

We thank the three reviewers for their thoughtful consideration of this manuscript and feel the changes suggested improve the clarity of our results. We detail the changes made below.

Reviewer 1:

1. I have one minor concern with the model proposed by the authors and their interpretations. *Sox2* is not simply restricted to the prosensory domain of the cochlear duct, but remains expressed in at least some cells of Kölliker's organ closer to the inner hair cell region. This domain of *Sox2*+ KO cells likely overlaps with *Prdm16* expression. *Sox2* is down-regulated from this region in a basal-apical gradient in the first few days after birth, but is certainly present in parts of KO at late embryonic ages. These cells do not normally give rise to hair cells and supporting cells and are not therefore "prosensory". How do the authors reconcile this with their model? Might EBF1 be repressing prosensory formation and hair cell and supporting cell differentiation independently of its repression of *Sox2*? If they examine their immunostaining closely, how many cells express both EBF1 and *Sox2*, and for how long between E14 and birth?

We agree with the reviewer's comments and have modified the text to address these points (lines 312-328). Although *SOX2* is one of the best characterized prosensory markers of the cochlea (Dabdoub et al., 2008; Kiernan et al., 2005), there is weak expression at the lateral edge of the Kölliker's organ. We have updated our Discussion to acknowledge that *Sox2* is not uniformly downregulated in the Kölliker's organ and propose that additional factors likely help maintain weak *SOX2* expression in the region closer to the inner hair cells.

Our multiome analyses point to a strong relationship between EBF1 and *SOX2*. For one, loss of EBF1 leads to an overall increase in chromatin accessibility within the Kölliker's organ (Fig. 4A), and

these regions are enriched with SOX2 binding motifs. Also, we see evidence that PRDM16, a proposed direct target of EBF1, directly regulates *Sox2* expression (Fig. 4E).

Although we did not quantify the number of cells that express both EBF1 and SOX2, we explored the dynamic expression profiles of EBFs and SOX2 more in our 2024 publication. This work includes anti-SOX2 and anti-EBF immunostaining experiments performed at E12.5, E14.5, E16.5, and E18.5. We found that EBF and SOX2 expression overlap at the medial edge of the cochlear floor at E12.5, and starting around E14.5, strong SOX2 expression shifts to the developing sensory epithelium while strong EBF expression persists in the Kölliker's organ.

2. One more minor point - the acknowledge thank a colleague for helping with "optimizing the protocol used to snap freeze the embryonic cochlear ducts in liquid nitrogen". Since it would appear this protocol is critical to the success of the project, it would be helpful if the authors described this procedure in some detail.

We have provided more detail on the protocols used for freezing cochlear ducts in liquid nitrogen, thawing the samples, and isolating their nuclei (lines 525-527 and 531-538).

Reviewer 2:

1. Furthermore, most of the proposed regulatory relationships, except for *Ccnjl*, have previously been identified or predicted.

The reviewer is correct that previous studies have proposed regulatory relationships between PRDM16 and *Jag1/Sox2* (Ebeid et al., 2022; Zhang et al., 2025), as well as between EBF1 and *Jag1/Prdm16/Sox2* (Kagoshima et al., 2024; Powers et al., 2024). However, we are the first to assess these relationships in the developing cochlea with multiome sequencing. This approach gave us a comprehensive view of EBF1's interactions within a regulatory network, allowing us to infer direct regulation of targets by EBF1 and predict possible roles for other transcription factors within its network.

2. Multiome analysis may imply, but does not prove, mechanism. It was used to identify candidate targets of Ebf1 regulation however, no follow-up experiments were performed to rigorously address the authors' claims that *Prdm1* and *Ccnjl* are direct targets of Ebf1. Consequently, the results are correlative and require functional studies to validate these claims. For example, is *Prdm1* bound to the sites in question and is this recruitment required for target gene regulation?

We have updated the wording throughout our manuscript to make it clearer these are proposed transcriptional targets. We also revised the Study Limitations section of our Discussion to emphasize that multiome sequencing helped mitigate the confounding noise of whole tissue analysis and provided a more comprehensive view of EBF1's role within a regulatory network. In addition to identifying EBF1's direct regulation of *Prdm16*, *Jag1*, and *Ccnjl*, we were able to detect PRDM16's direct regulation of *Jag1* and *Sox2*, as well as SOX2's direct regulation of *Ccnjl*. We acknowledge that ChIP-seq, CUT&RUN, or CUT&Tag would need to be performed to confirm direct binding of EBF1 to the proposed direct targets. However, these epigenomic profiling techniques work best with a primary antibody specific to the protein of interest. Our lab found that commercially available antibodies that recognize mouse EBF1 worked poorly, if at all, and demonstrated high background in our immunostaining experiments. The identification of EBF1-specific binding within the cochlear epithelium is further confounded by the transcription factor's high expression levels in the surrounding mesenchyme in addition to the epithelium (Fig. 2A and 3A). Even with dissections aimed to enrich for cochlear epithelium, we still capture a fair amount of mesenchyme. In our E14.5 datasets, 29.09% of the nuclei are from mesenchymal cells and 53.46% are from epithelial cells (Fig. S1). In our E16.5 datasets, 44.56% of the nuclei are from mesenchymal cells and 41.27% are from epithelial cells (Fig. S4). In short, we chose multiome sequencing for a more global analysis and because analysis of EBF1-specific binding in the cochlear epithelium is complicated by a lack of a high-quality EBF1-specific antibody and EBF1 expression outside of the cochlear epithelium.

3. How does the weaker ATAC-seq peak at the *Ccnjl* promoter in *Ebf1cko* factor into the authors'

model?

We have excluded the ATAC peak at the promoter from our model because it does not correlate with *Ccnjl* expression based on our linkage analysis comparing peak accessibility and gene expression across cells.

4. In between the rows of iHCs and oHCs are two rows of pillar cells (inner and outer) that differentiate with actin rich structures to form the tunnel of Corti (Fig. 1). The aforementioned statement is inaccurate and should be amended as follows: Pillar cells are enriched with microtubule arrays that are maintained through actin crosslinks.

We have updated this statement in our Introduction (lines 53-56).

5. The sensory epithelium (SE) lies above the basilar membrane which vibrates in response to sound, allowing the stereocilia on the tops of the HCs to deflect and thereby causing neurotransmitter release from the iHCs. The aforementioned statement lacks an important detail and should be amended as follows: The sensory epithelium (SE) lies above the basilar membrane, which vibrates in response to sound. This vibration causes deflection of the stereocilia on the tops of the HCs, initiating mechanotransduction and leading to neurotransmitter release from the IHCs.

We have updated this statement in our Introduction (lines 56-58).

Reviewer 3:

1. The authors indicate in the abstract and main text that EBF1 may promote cell cycle exit by repressing *Ccnjl* expression. However, no functional evidence is provided to support this. Unlike other identified *Ebf1* targets such as *Prdm16*, *Jag1*, and *Sox2*, which have established roles in auditory sensory development, the function of *Ccnjl* in cochlear development has yet to be established. To remedy this the authors could use an in vitro approach such as cochlear explants and overexpress *Ccnjl* (or overexpress/ knockdown other candidates) and evaluate its effect on cell proliferation.

We agree that the functional testing of CCNJL's role in the cochlea would be the best way to validate this as a critical target; however, it would be difficult to test this in vitro, since we would likely have to introduce/knockdown the gene very early in development to replicate the phenotype from *Ebf1*-cKO mice. The ideal way to test for rescue would be to generate a mouse knockout line for *Ccnjl* and cross this to the *Ebf1*-cKO, but we feel this would be beyond the scope of the current study.

2. The main text should specify the Cre strategy used for the conditional knockout of *Ebf1*.

Information on the Cre strategy has been added to the Introduction (lines 90-92).

3. Readme files for the supplemental tables should be included.

We have updated the supplemental table descriptions with additional details.

References:

- Dabdoub, A., Puligilla, C., Jones, J. M., Fritzscht, B., Cheah, K. S. E., Pevny, L. H. and Kelley, M. W. (2008). *Sox2* signaling in prosensory domain specification and subsequent hair cell differentiation in the developing cochlea. *Proc. Natl. Acad. Sci. USA* 105, 18396-18401. doi:10.1073/pnas.0808175105
- Ebeid, M., Barnas, K., Zhang, H., Yaghmour, A., Noreikaite, G. and Bjork, B. C. (2022). PRDM16 expression and function in mammalian cochlear development. *Dev. Dyn.* 251, 1666-1683. doi:10.1002/dvdy.480
- Kagoshima, H., Ohnishi, H., Yamamoto, R., Yasumoto, A., Tona, Y., Nakagawa, T., Omori, K. and Yamamoto, N. (2024). EBF1 limits the numbers of cochlear hair and supporting cells and forms the scala tympani and spiral limbus during inner ear development. *J. Neurosci.* 44, e1060232023. doi:10.1523/JNEUROSCI.1060-23.2023

- Kiernan, A. E., Pelling, A. L., Leung, K. K. H., Tang, A. S. P., Bell, D. M., Tease, C., Lovell- Badge, R., Steel, K. P. and Cheah, K. S. E. (2005). Sox2 is required for sensory organ development in the mammalian inner ear. *Nature* 434, 1031-1035. doi:10.1038/nature03487
- Powers, K. G., Wilkerson, B. A., Beach, K. E., Seo, S. S., Rodriguez, J. S., Baxter, A. N., Hunter, S. E. and Bermingham-McDonogh, O. (2024). Deletion of the *Ebf1*, a mouse deafness gene, causes a dramatic increase in hair cells and support cells of the organ of Corti. *Development* 151. doi:10.1242/dev.202816
- Zhang, H., Papiernik, T., Tian, S., Yaghmour, A., Alzein, A., Lennon, J. B., Maini, R., Tan, X., Niazi, A., Park, J., et al. (2025). Kölliker's Organ Functions as a Developmental Hub in Mouse Cochlea Regulating Spiral Limbus and Tectorial Membrane Development. *J. Neurosci.* 45, e0721242025. doi:10.1523/JNEUROSCI.0721-24.2025

Second decision letter

MS ID#: dev.205207R1

MS TITLE: EBF1 regulates sensory establishment in the cochlea by positioning the medial boundary of the prosensory domain and restricting proliferation of the sensory progenitor population

AUTHORS: Kathryn G. Powers, Joshua Hahn, Juliette Wohlschlegel and Olivia Bermingham-McDonogh

Dear Dr Bermingham-McDonogh,

I am happy to tell you that your manuscript has been accepted for publication in *Development*, pending our standard publication integrity checks.

Reviewer 1

Advance summary and potential significance to field

This manuscript is a follow up to a recent paper from the Bermingham McDonogh lab on the role of the EBF1 transcription factor in the development of the cochlea. This tissue has great interest for developmental biologists due to its highly organized cellular structure where the numbers of its constituent cell types are tightly regulated. Moreover, the mammalian cochlea exhibits a remarkable decoupling of cell cycle exit from differentiation, whereby cells in the apical tip of the cochlea (that will ultimately detect low frequency sounds) exit the cell cycle first but then wait for four or five days before differentiating. In contrast, cells at the base (high frequency) region of the cochlea exit the cell cycle last, but almost immediately differentiate.

The authors previously showed that inactivation of the EBF1 gene led to a great over-production of hair cells and supporting cells in the cochlea, and that this over-production is due in part to a significant expansion of the Sox2+ prosensory domain from which these cells differentiate. The present study seeks a mechanistic explanation for these results by performing multiomic single cell analysis of the embryonic cochlea duct at two ages. They propose model in which EBF1 promotes expression of the transcription factor Prdm16 in the medial side of the cochlear duct. This gene in turn represses Sox2, a marker of the prosensory domain of the cochlear duct. In EBF1 knockouts, Prdm16 is largely lost, leading to an up-regulation of Sox2, an expansion of the prosensory domain and increased numbers of hair cells and supporting cells. They also provide evidence for EBF1 regulating a cell cycle component, CCNJL.

Comments for the author

The authors have addressed my previous comments and I have no more concerns with the manuscript

Reviewer 2

Advance summary and potential significance to field

I appreciate the authors' response to my critiques, but it does not resolve my principal concern that the multiome analysis fails to provide additional mechanistic insight into the Ebf1 gene regulatory network beyond what was reported in their previous publication (Powers et al., 2024). In the absence of follow up experiments to test the functional significance of the ATAC seq peaks and putative binding motifs, the data remain highly correlative and lack the rigorous investigation needed to substantiate their proposed model.

Reviewer 3

Advance summary and potential significance to field

The study by Powers et al. provides new insights into the gene regulatory network governing the position of the medial boundary between the auditory sensory epithelium and a group of transient epithelial cells termed Kölliker's cells. Defects in the positioning of this boundary alter auditory development and lead to hearing loss. The authors recently demonstrated that early otic loss of the transcription factor EBF1 results in a significant medial expansion of the auditory sensory domain, accompanied by ectopic cell proliferation. To uncover the underlying mechanisms, the authors conducted multiomic experiments using cochlear tissue from E14.5 and E16.5. By integrating single-nucleus RNA sequencing and ATAC sequencing data, they show that EBF1 prevents Kölliker's cells from adopting a sensory fate by activating Prmd16 expression and repressing Jag1 and Sox2 expression. Additionally, they present tentative evidence that EBF1 may promote cell-cycle exit in the developing cochlear epithelium by suppressing Ccnjl expression.

Comments for the author

No further suggestions. The provided revisions to the study are satisfactory.